# Obstructive Sleep Apnea Syndrome In Vitro Model: Controlled Intermittent Hypoxia Stimulation of Human Stem Cells-Derived Cardiomyocytes

**DOI:** 10.3390/ijms231810272

**Published:** 2022-09-07

**Authors:** Danielle Regev, Sharon Etzion, Hen Haddad, Jacob Gopas, Aviv Goldbart

**Affiliations:** 1Shraga Segal Department of Microbiology, Immunology and Genetics, Faculty of Health Sciences, Ben-Gurion University of the Negev, Beer-Sheva 8410501, Israel; 2Regenerative Medicine & Stem Cell Research Center, Ben-Gurion University of the Negev, Beer-Sheva 8410501, Israel; 3Soroka University Medical Center, Department of Oncology, Ben-Gurion University of the Negev, Beer-Sheva 8410501, Israel; 4Soroka University Medical Center, Department of Pediatrics, Faculty of Health Sciences, Ben-Gurion University of the Negev, Beer-Sheva 8410501, Israel; 5Pediatric Pulmonary and Sleep Research Laboratory, Faculty of Health Sciences, Ben-Gurion University of the Negev, Beer-Sheva 8410501, Israel

**Keywords:** obstructive sleep apnea, intermittent hypoxia, inflammation, NF-κB, cytokines, hESC-CM, human embryonic stem cells derived cardiomyocytes

## Abstract

Cardiovascular morbidity is the leading cause of death of obstructive sleep apnea (OSA) syndrome patients. Nocturnal airway obstruction is associated with intermittent hypoxia (IH). In our previous work with cell lines, incubation with sera from OSA patients induced changes in cell morphology, NF-κB activation and decreased viability. A decrease in beating rate, contraction amplitude and a reduction in intracellular calcium signaling was also observed in human cardiomyocytes differentiated from human embryonic stem cells (hESC-CMs). We expanded these observations using a new controlled IH in vitro system on beating hESC-CMs. The Oxy-Cycler system was programed to generate IH cycles. Following IH, we detected the activation of Hif-1α as an indicator of hypoxia and nuclear NF-κB p65 and p50 subunits, representing pro-inflammatory activity. We also detected the secretion of inflammatory cytokines, such as MIF, PAI-1, MCP-1 and CXCL1, and demonstrated a decrease in beating rate of hESC-CMs following IH. IH induces the co-activation of inflammatory features together with cardiomyocyte alterations which are consistent with myocardial damage in OSA. This study provides an innovative approach for in vitro studies of OSA cardiovascular morbidity and supports the search for new pharmacological agents and molecular targets to improve diagnosis and treatment of patients.

## 1. Introduction

Obstructive sleep apnea (OSA) syndrome is the most common sleep-related breathing disorder. It is estimated to affect 15–24% of adults, but it is greatly underdiagnosed [1,2]. It is characterized by a collapse of the pharynx during sleep and repetitive nocturnal upper airway obstructive events associated with intermittent hypoxia (IH) [3]. OSA has a wide array of morbidities such as neurocognitive, pulmonary, and cardiovascular damage. Cardiovascular morbidity is considered to be the leading cause of death from OSA and encompasses many morbidities such as hypertension, coronary artery disease, heart failure, arrhythmia, ventricular hypertrophy and myocardial ischemia [4,5].

HIF-1 (hypoxia inducible factor-1) is a transcription factor that has a key role in cell adaption to hypoxic conditions [6]. In normoxic conditions, the Hif-1α subunit is ubiquitinated and degraded by the proteosome. In hypoxic conditions, Hif-1α is stable and is translocated to the nucleus where it dimerizes with the Hif-1β subunit, thus creating a complex that can activate the transcription of genes involved in metabolic cellular adaption to hypoxia and inflammation [7]. IH activates the HIF-1 signaling pathway which participates in OSA morbidity and cardiovascular diseases [8,9].

Nuclear factor kappa-light-chain-enhancer of activated B cells (NF-κB) is a transcription factor that has a major role in inflammation by regulating the expression of many cytokines and chemokines and has been linked to OSA as well as to cardiovascular diseases [10].

Our laboratory has previously shown that there is a local NF-κB activation and IL-1α expression in adenoid and tonsillar tissue of children with OSA [11], as well as activation of NF-κB in cells incubated with sera of these children [12]. We also were able to show for the first time the effect of sera from OSA patients on beating human cardiomyocytes differentiated from human embryonic stem cells (hESC-CMs). We showed NF-κB activation as well as a decrease in beating rate, contraction amplitude and a decrease in calcium signaling [13].

Currently, there is no good solution to the cardiovascular sequel that occurs in OSA treated by continuous positive airway pressure (CPAP). Our study looks at human cardiomyocytes under hypoxic conditions and aims to identify mechanisms that may become a target for intervention. We search for those mechanisms at the molecular, cellular and physiological levels.

The conventional treatment of OSA is by CPAP. Although CPAP restores respiration and sleep architecture, several randomized controlled trials (RCTs) and meta-analyses have reported no risk reduction in adverse CV events from the use of CPAP therapy in OSA it does not reduce cardiovascular morbidity [14,15,16,17].

Considering IH as a key feature of OSA and activation of the NF-κB pro-inflammatory pathway [10] it is of much interest to understand the interplay of these factors and their effect on cardiac damage in OSA.

In order to study the detrimental effect of cyclical IH on cardiomyocytes, we established a unique in vitro system not previously utilized in OSA research. Our system, mimics the pattern of IH in patients on hESC-CMs. We measured different biochemical and cellular parameters as compared to these cells under conditions of normoxia. By understanding the mechanism(s) of cardiovascular morbidity in OSA, we may be able to detect new biomarkers as targets for drug development to diagnose and assess the severity of the disease and to determine treatment efficiency and the extent of cardiac damage.

## 2. Results

### 2.1. Hif-1α Expression Is Increased in hESC-CMs following IH

Hif-1α protein expression is upregulated in serum of OSA patients both in the evening and morning [18]. IH is a key component of OSA and hypoxia increase Hif-1α expression. Hif-1α is expressed in normoxic and hypoxic conditions [19]. Here, we determined the expression of Hif-1α in hESC-CMs following IH or normoxic conditions by immunofluorescence using the Operetta system (Figure 1). The IH protocol was optimized after several O_2_ concentrations were tested and was based on similar O_2_ concentrations in other in vitro systems [20]. In all the experiments, O_2_ levels alternated between 1% for 8 min and ~21% for 4 min (see details in Materials and Methods, Section 4.3). Hif-1α expression was detected in the nucleus both in cells under IH and normoxic conditions, its nuclear expression was significantly upregulated in cells under IH conditions and in DFO and CoCl_2_-treated, positive control cells (Figure 1A).

These results confirm the ability of this system to effectively induce and detect intermittent hypoxia.

### 2.2. Decrease in hESC-CMs Beating Rate following IH

Clinical research showed a decrease in beating rate during apneic episodes in OSA patients [3]. We previously showed a decrease in beating rate following incubation with sera from OSA patients [13]. Here we asked whether IH conditions will affect the beating rate of hESC-CMs. The results (Figure 2) show a significant reduction in beating rate following IH compared to cells under normoxia. Movies of hESC-CMs beating rate in normoxia and IH are shown in the Appendix A.

### 2.3. Increased NF-κB Activation following IH in hESC-CMs

Based on our previous results where we showed NF-κB activation in tonsils of OSA patients [11] and in CMs incubated with sera from patients [12], we asked whether IH will activate the NF-κB pathway as well. We determined the nuclear expression of NF-κB subunits p65 and p50 following normoxia or IH by immunofluorescence using the Operetta system (Figure 3).

The results show an increase in nuclear expression of both p65 and p50 subunits following IH (Figure 3A), including a distinct p65 perinuclear staining (Figure 3C, yellow arrow).

### 2.4. Detection of Cytokines following IH in hESC-CMs

OSA patients are predisposed to cardiovascular diseases by several mechanisms; one of them is IH-induced inflammation. Pro-inflammatory proteins are present in OSA patients’ sera and are released from CMs [21,22]. These results prompted us to ask whether hESC-CMs produce cytokines to the medium and how IH affects their production.

We collected supernatants following 12 h of normoxia or immediately after 12 h of IH. Supernatants were also collected from cells under 12 h of IH followed by 24 h of normoxia.

We detected four cytokines (out of a panel of 36): MIF (macrophage migration inhibitory factor), PAI-1 (plasminogen activator inhibitor-1), MCP-1 (monocyte chemoattractant protein-1) and CXCL1 (chemokine C-X-C motif ligand-1).

MIF and PAI-1 were detected in normoxic cells (Figure 4A) and elevated following 12 h of IH (Figure 4B). MIF was also elevated in cells initially exposed to IH followed by 24 h of normoxia (Figure 4D) but not in the normoxic cells (Figure 4C). PAI-1 relative amounts were high and similar after 24 h of normoxia between cells initially under normoxia or IH. (Figure 4A–D). MCP-1 and CXCL1 were not detected after 12 h but were noticed after an additional 24 h of normoxia with a higher level in cells initially exposed to IH (Figure 4C,D).

## 3. Discussion

We have previously shown NF-κB activation in adenoids and tonsils tissues of OSA patients [11]. We also showed NF-κB activation in neonatal rat cardiomyocytes and human immortalized cardiomyocytes following incubation with sera of OSA patients. Exposure to sera also showed morphological changes, decreased viability and contractility in these cells [12]. In order to examine the model in human cells, we chose to use human embryonic stem cells derived cardiomyocytes with contractile capabilities. We incubated the cells with sera of OSA patients and showed NF-κB activation, a decrease in beating rate as well as a decrease in contraction amplitude and calcium signaling [13].

Our previous results strengthen the notion that there is a major inflammatory pathway that negatively impacts the cardiovascular system among OSA patients. Therefore, we further investigated similar parameters as detected with sera incubation in a novel system, mimicking the intermittent hypoxia that defines OSA patients’ sleep.

Since IH is a key feature of OSA which also activates the NF-κB pro-inflammatory pathway [10], we hypothesized that cyclical IH has a negative effect on cardiomyocytes in several dimensions. In order to study this feature, we established an in vitro system not previously utilized in OSA research. We studied the effect of cyclic IH through the OxyCycler incubator, mimicking the pattern of IH in patients, on hESC-CMs.

We first sought to establish the ability of our system to induce IH and whether CMs respond to it. We measured the expression of nuclear Hif-1α, a known hypoxia marker that is also upregulated in OSA patients [18]. Following IH, we observed an increase in Hif-1α nuclear expression in CMs when compared to normoxia. These results validated the system and enabled us to apply it to study the effect of IH on CMs.

OSA patients display cardiac dysfunction which include a decrease in beating rate during apneic episodes [3]. Previous work in our laboratory showed a decrease in beating rate following incubation of CMs with sera from patients. Here we measured the beating rate of CMs before and after exposure to IH. Similarly, for incubation of the cells with OSA sera, we showed a significant decrease in beating rate following IH. Cells kept under normoxic conditions over time did not show a significant reduction in beating rate. As seen in the films in Appendix A the decrease in beating rate is also accompanied by an observed decrease in the contraction strength in cells under IH. This effect is reversible by fresh medium replacement and returning the cells to normoxic conditions (not shown). The results thus suggest that protein effectors such as cytokines and/or small molecules released by the cells during IH may be involved, potentially pointing to the initial processes that promote cardiac dysfunction in OSA patients [14]. Although sleep is normally a time when parasympathetic modulation of the heart predominates and myocardial electrical stability is enhanced, OSA and CSA disturb this quiescence, creating an autonomic profile in which both profound vagal activity, leading to bradyarrhythmias, and symptho-excitation favouring ventricular ectopy are observed. The resulting tendency toward cardiac arrhythmia may directly contribute to sudden cardiac death and premature mortality in patients with sleep apnea [23,24]. Thus, since both a decrease or an increase in beating rate is observed in OSA patients, either variation from the norm, including during apneic episodes can be of significance and important in promoting arrythmias.

NF-κB pro-inflammatory effect is a key feature in OSA and is known to have a role in cardiac damage [9,10]. We measured the nuclear expression of NF-κB subunits p65 and p50 following IH. We observed a significant increase in nuclear expression of both p65 and p50. Perinuclear staining of p65 has been mentioned in the context of NF-κB activation in several cell types and as a specific phenomenon to CMs rather than cardiofibroblasts [25,26,27]. Perinuclear localization is thought to play a role in NF-κB activation since it occurs during activation and nuclear translocation of p65 and is decreased when NF-κB is inhibited [27]. It is involved in cardiac pathogenesis and has been reported to induce cardiac damage, myocardial infarction [27] and ventricular hypertrophy [28].

These results are consistent with our previous results in tonsils of OSA patients [11] and cell lines following incubation with OSA serum [12]. Adult OSA patients also show an increase in activated NF-kB [29].

NF-κB signaling pathway promotes inflammation by inducing expression of pro-inflammatory cytokines. It has been shown that IH plays a role in OSA pathogenesis and cardiovascular complications through activation of pro-inflammatory pathways [6,30].

It has also been reported that CMs secrete a variety of molecules (“cardiokines”) that are involved in physiological and pathological processes [21]. We measured the presence of cytokines and chemokines in the medium of CMs following 12 h of normoxia or immediately after 12 h of IH. Supernatants were also collected from cells that underwent 12 h of IH followed by 24 h of normoxia.

Interestingly, a recent paper by Díaz-García et al. [31] and the comment on this article by Borker and Patel [32] refer to the mechanism that links OSA with the regulation of the systemic inflammatory response, the activation of the inflammasome, in particular the nucleotide-binding oligomerization domain-like receptor 3 (NLRP3). They analyzed NLRP3 activity in monocytes and plasma of patients with severe OSA. In addition, they confirmed similar results in in vitro conditions of monocytes under their IH protocol. The authors conclude that NLRP3 activation triggering inflammatory cytokines such as IL-1β and IL-18 might be a linking mechanism between IH and other OSA-induced immediate changes with the development of a systemic inflammatory response.

These results support our previous and current work since priming of the inflammasome pathway is triggered by activation of NF-κB signaling and production of pro-inflammatory cytokines.

Similar to our results where we showed that Hif-1α increases on CMs upon IH, the authors also demonstrate that Hif-1α is upregulated in monocytes of patients with OSA and that ex vivo inhibition of Hif-1α led to a reduction in inflammasome transcription, suggesting that Hif-1α may play a prominent role in the priming step.

Ex vivo exposure of monocytes from healthy participants to a severe IH paradigm increased priming but was not sufficient to activate the inflammasome complex. The addition of plasma from patients with OSA to the IH exposure, however, increased both priming and activation, suggesting the presence of an activation signal circulating in patients with OSA. These results further support our previous observations that sera from pediatric OSA patients provide a strong pro-inflammatory signal. It would be of interest to investigate the activation of NLRP3 under IH on human CMs.

We detected 4 cytokines out of 36: MIF, PAI-1, MCP-1 and CXCL1.

MIF and PAI-1 were detected both in normoxic and IH cells and at both time points. They were elevated under IH after 12 h. MIF was also elevated after 24 h normoxia and more elevated in cells after initial IH conditions. PAI-1 relative amounts were high and similar after 24 h between cells initially under normoxia or IH. MCP-1 and CXCL1 were not detected after 12 h but were significantly elevated following IH after 24 h.

MIF is a circulating cytokine that can be expressed by many cell types as well as cardiomyocytes. MIF is stored in intracellular pools and is released as a result of pro-inflammatory stimuli, oxidative stress, ischemia-reperfusion and hypoxia [33,34]. In addition to its role in inflammation, MIF had been linked to OSA [35], and cardiovascular diseases such as atherosclerosis, myocardial infarction, coronary heart diseases and hypertrophy [33,36,37]. Our results show an increase in MIF secretion following IH which remains higher after additional 24 h in normoxic conditions. MIF has a dual role in CMs following hypoxia; in the short term, it is cardioprotective, but in the long-term MIF has detrimental consequences [38].

PAI-1 is a profibrotic factor; it is expressed in many cell types including CMs. It can be induced by growth factors, NF-κB induced inflammatory cytokines and IH [39,40]. PAI-1 profibrotic properties are also associated with atherosclerosis, fibrosis, OSA and cardiovascular diseases [41,42]. PAI-1 is induced both by NF-κB and Hif-1α [41,42].

MCP-1 is a chemokine which is expressed by several cell types such as vascular endothelial cells, monocytes, fibroblasts and CMs [43,44]. MCP-1 is upregulated following NF-κB pro-inflammatory stimuli and hypoxia [26,27,45]. The association of MCP-1 with hypoxia and inflammation is related to OSA and cardiovascular diseases [44,45].

CXCL1 is an inflammatory chemokine and can be expressed by neutrophils, macrophages, epithelial cells, cardiofibroblasts and CMs [26,27,46]. It is secreted following NF-κB activation, pro-inflammatory cytokines and neuro-humoral factors such as angiotensin (Ang) II [47]. CXCL1 has chemoattractant properties that play a role in the injured heart by promoting inflammation and fibrosis. It was shown to be upregulated in myocardial infarction, ischaemia/reperfusion injury, atherosclerosis, hypertension, heart failure and hypertrophy [46,47]. Although CXCL1 is expressed following NF-κB activation, hypoxia and inflammation, to our knowledge, it has not been reported to be associated with OSA.

Our in vitro model reflects at the cellular level the effects of IH and inflammation. However, this system, although innovative in its approach, also has obvious limitations common to other in vitro models, such as the lack of physiological extracellular factors affecting the tissue and the fact that the cardiomyocytes are young and do not differentiate to mature cells forming a complete organ. In addition, the pathology in patients is a result of long term (years) cumulative effects of OSA. In contrast, here we study the effects in the short time span of hours/days. Therefore, we are careful in our conclusions as pertinent to the disease. Having said that, we believe that the results obtained using human embryonic stem cells derived cardiomyocytes enable us to better understand the effect of OSA on heart cells at the molecular, cellular and physiological levels. Understanding the effect of IH in terms of signaling pathway activation and functionality may shed light to the understanding of cumulative damage encountered by the heart in this prevalent condition.

## 4. Materials and Methods

### 4.1. Antibodies

Antibodies Anti-NF-κB p65 NLS specific [600-401-271]: Rockland Immunochemicals Inc., Limerick, PA, USA.

Anti NFκB p50 (NLS) [sc-114]: Santa Cruz Biotechnology, Dallas, TX, USA.

Rabbit anti NFκB p65 622602: BioLegend, San Diego, CA, USA.

Rabbit anti NFκB p50 [AHP2331]: Bio-Rad Laboratories, Hercules, CA, USA.

Anti-Cardiac Troponin T antibody [1C11], Abcam, Cambridge, UK.

Goat anti-Mouse IgG (H+L) Highly Cross-Adsorbed Secondary Antibody, Alexa Fluor

488 [A-11029] and Goat anti-Rabbit IgG (H+L) Highly Cross-Adsorbed Secondary Antibody, Alexa Fluor 633 [A-21071], Invitrogen, Thermo Fisher Scientific, Waltham, MA, USA.

Vectashield Mounting Medium with DAPI: Vector Laboratories, Burlingame, CA, USA.

### 4.2. Cardiomyocytes (CMs) Differentiation

WA-09, human embryonic stem cells were obtained from Dr. Rivki Ofir and differentiated to cardiomyocytes as described before [14,45]. Briefly, hES cells were maintained on Matrigel in NutriStem medium, dissociated into monocultures cells with Accutase solution [46] at 37 °C, for 2 min and seeded to a density of 8.5 × 105 cells/well (in 12-well plates). The amount of Matrigel coating was doubled as recommended by the manufacturer. The cells were cultured in NutriStem supplemented with the ROCK inhibitor, Y-27632 (5-μM) for 1 day (day −5), then in NutriStem medium, which was changed daily. After 5 days, at day 0, the cells were supplemented with the Gsk3 inhibitor CHIR99021 (6 μM) in RPMI that contains 1:50 B27 without insulin for 1 day. On day 1, the medium was removed and replaced with fresh medium and Gsk3 inhibitor for 1 day. On day 2 the medium was changed to RPMI/B27 without insulin for another 1 day. Next (day 3), the Wnt production-2 inhibitor, IWP2 (3 μM), was added for an additional 1 day and then was removed (day 4) by changing the medium to RPMI/B27 without insulin. The medium was changed every day, but not on day 6. Finally, on day 7, the cells were maintained in RPMI/B27 (1:50) supplement medium that was changed every day. First, spontaneous cell beatings appeared on days 7 to 11 of differentiation. The cells were kept in 5% CO_2_ in a 37 °C heated incubator. Cells were transferred to 96 wells (50 × 10^3^ cells/well) for evaluation of different parameters.

### 4.3. Hypoxia Induction by Using the “OxyCycler System”

Intermittent hypoxia (IH) exposure was conducted using a custom-designed computer-controlled incubator chamber connected to BioSpherix OxyCycler (Biospherix), Redfield, NY, USA). Cells were maintained at 37 °C at 5% CO_2_ in the hypoxic chamber in which O_2_ levels were alternated between 1% for 8 min and ~21% for 4 min (Figure 5). Cells in the control group were maintained in normoxic conditions (~21% O_2_ and 5% CO_2_) throughout the experiment. Cells were exposed to normoxia or intermittent hypoxia for 12 h, a total of 60 cycles in 12 h.

### 4.4. Beating Rate Measurement

Beating rate was determined initially on cells under normoxia at time 0 and immediately after 12 h of IH or normoxic conditions. Beating cells in different areas of the well were scored for the number of cell-contractions/minute under the microscope. A total of 10 contracting cell aggregates were counted/well/minute; 15 wells were scored per experiment. Each experiment was repeated three times. Statistical significance was determined.

### 4.5. Operetta High-Content Imaging System—PerkinElmer

In order to quantify the expression of Hif-1α and activated NF-κB in the nuclei of cardiomyocytes (troponin positive cells), we quantified the relative amount of the specific protein by immunofluorescence. A total of 50,000 differentiated CM cells were plated per well of a 96 well plate and allowed to attach for 24 h at 37 °C. Then the cells were transferred to an OxyCycler incubator. Cells were maintained at 37 °C at 5% CO_2_ in the hypoxic chamber in which O_2_ levels were alternated between 1% for 8 min and ~21% for 4 min, a total of 60 cycles in 12 h. Cells in the control group were maintained in normoxic conditions (~21% O_2_ and 5% CO_2_) throughout the experiment. At the end of the experiment, the cells were washed twice with PBS and fixed with 4% paraformaldehyde in PBS at room temperature (RT) for 20 min. Cells were washed twice again in PBS before permeabilization in 3% FBS in PBS and blocked with 0.1% Triton X100 in 3% FBS and PBS for 60 min at RT. Cells were washed twice with 3% FBS in PBS before staining with an antibody to cardiac Troponin T (cTnT), a known CM marker troponin [48], and the desired primary antibody (p65, p50 etc.) in 3% FBS and 0.1% Triton X100 in PBS for overnight. The cells were then stained with a secondary antibody AF488 (green) and AF 688 (red) for overnight and DAPI (blue-nuclear) staining for 30 min, followed by 3× washing with 0.1% Triton X100 in 3% FBS and PBS, cells were imaged with the Operetta high-content imaging system at 40× magnification. Analysis of the pictures was performed through the Columbus server of the company where we can define parameters such as selective fluorescence quantitative scoring of cardiomyocytes (green cells), and the antigen of interest in the nucleus or the cytoplasm (pink) specifically in these cells.

As positive controls for the expression of Hif-1α, we pretreated the cells with deferoxamine mesylate (DFO, 100 µM) or cobalt chloride (CoCl2, 100 µM), which induces chemical hypoxia [49].

### 4.6. Determination of Specific Proteins in Supernatants

Determination of various cytokines, chemokines and growth factors in supernatants from cardiomyocytes under normoxia or intermittent hypoxia was performed by the proteome profiler array, human cytokine array Cat. ARY005B, R&D systems Inc. Densitometry was performed with an Azure c300 Gel Imaging System by Azure Biosystems and analyzed with the ImageJ software version 1.53e, https://imagej.nih.gov/ij/ 24 March 2022. In the membranes, each protein was doted in duplicate. The duplicate densitometry values were averaged and compared to the average values obtained in the normoxia membrane immediately after IH or compared to normoxia after 24 h recovery. The results were expressed as a percentage of each protein compared to normoxia.

### 4.7. Statistical Analysis

Values were expressed as mean ± SEM. Statistical analysis was performed using Prism 7.0 (GraphPad Software, San Diego, CA, USA). Comparisons between normoxia and IH groups were performed using unpaired Student’s t-test or ANOVA analysis. The specific tests that were used are mentioned in the legend of each figure. The criteria for significance were set at *p* < 0.05. Unless otherwise stated, p-values were displayed graphically as follows: * *p* < 0.01, ** *p* < 0.001, *** *p* < 0.00015.

## Figures and Tables

**Figure 1 ijms-23-10272-f001:**
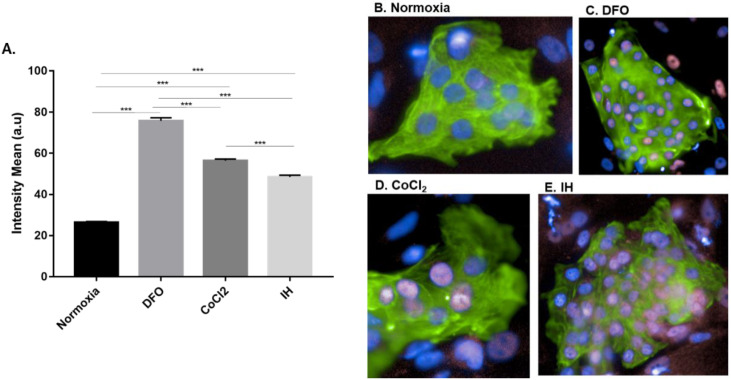
Increase in nuclear Hif-1α expression in hESC-CMs following IH: (**A**) quantification of the hypoxia marker Hif-1α expression in the nucleus following normoxia or IH only on CMs; normoxia = normal conditions (21% O_2_, 37 °C), DFO and CoCl_2_ were incubated in normal conditions as well; IH = intermittent hypoxia (1% O_2_, 37 °C) for 12 h (60 Cycles). Results are averages of 3 separate experiments performed in 10 replicates, ANOVA analysis *** *p* < 0.001, n = 37,661; (**B**) immunostaining of Hif-1α and troponin (cardiomyocytes specific marker) following normoxia or IH. Differentiated CMs were detected with antibodies to troponin, a marker of cardiomyocytes cells (green) and Hif-1α (pink), followed by the appropriate secondary antibody. Nuclei were stained with DAPI (blue); (**B**) normoxia conditions; (**C**) normoxia conditions and DFO; (**D**) normoxia conditions and CoCl_2;_ and (**E**) IH conditions; 40× magnification.

**Figure 2 ijms-23-10272-f002:**
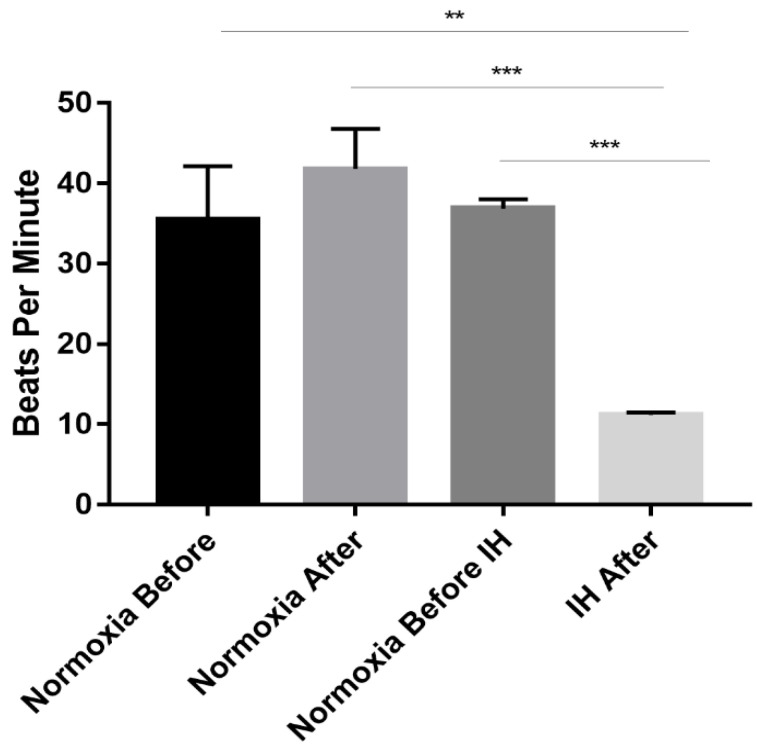
Decrease in hESC-CMs beating rate following IH. Beats per minute were determined on cells under normoxic conditions (21% O_2_, 37 °C) or intermittent hypoxia (IH) (1% O_2_, 37 °C) for 12 h (60 Cycles). Results are averages of 3 separate experiments. Each experiment was performed on 15 wells, in each well 10 contracting CMs cell aggregates were counted under the microscope. A total of 450 CMs in each group was scored. ANOVA analysis ** *p* = 0.001, *** *p* < 0.0001.

**Figure 3 ijms-23-10272-f003:**
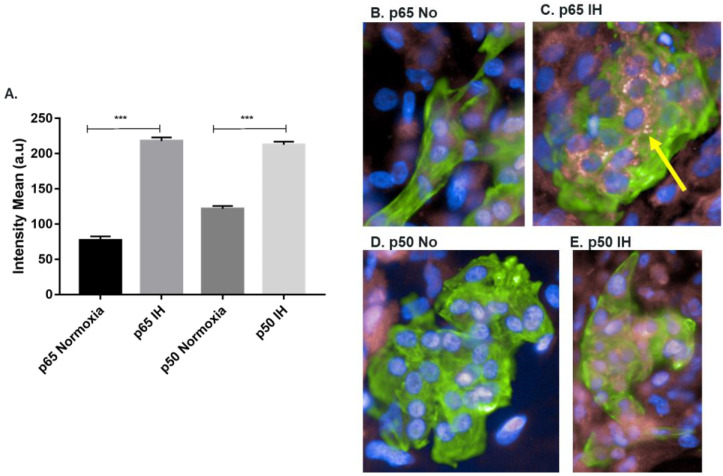
Increase in nuclear NF-κB sub-units p65 and p50 expression in hESC-CMs following IH: (**A**) quantification of the NF-κB subunits p65 and p50 expression in the nucleus following normoxia or IH only on CMs. Normoxia (21% O_2_, 37 °C), intermittent hypoxia (IH) (1% O_2_, 37 °C) for 12 h (60 Cycles). Results are averages of 3 separate experiments performed in 10 replicates Student’s t-test *** *p* < 0.001, n = 39,852; (**B**) immunostaining of p65, p50 and cardiac troponin T (cardiomyocytes specific marker) following normoxia or IH. Differentiated CMs were detected with antibodies to cardiac troponin T (green) and p65 or p50 (pink), followed by the appropriate secondary antibody. Nuclei were stained with DAPI (blue); (**B**) normoxia conditions and p65; (**C**) IH conditions and p65; (**D**) normoxia conditions and p50; and (**E**) IH conditions and p50. 40× magnification.

**Figure 4 ijms-23-10272-f004:**
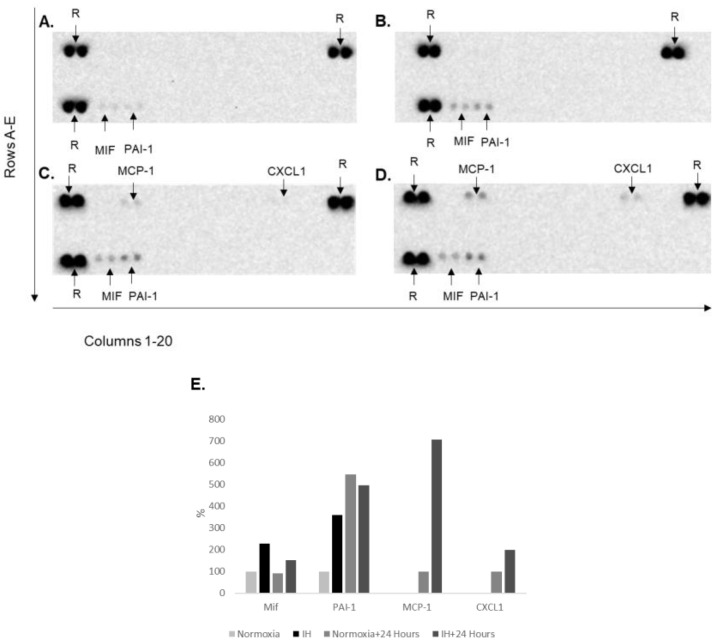
Specific protein secretion following IH in hESC-CMs. Supernatants following normoxia (21%O_2_, 37 °C) or IH = 1% O_2_, 37 °C for 12 h (60 Cycles) and 24 additional hours in normoxia conditions. R arrows are reference points. All the membranes were exposed to the same exposure parameters during densitometry: (**A**) membrane following 12 h normoxia; (**B**) membrane following 12 h intermittent hypoxia; (**C**) membrane following a total of 36 h normoxia; (**D**) membrane following 12 h intermittent hypoxia followed by 24 h normoxia; (**E**) imageJ densitometry analysis.

**Figure 5 ijms-23-10272-f005:**
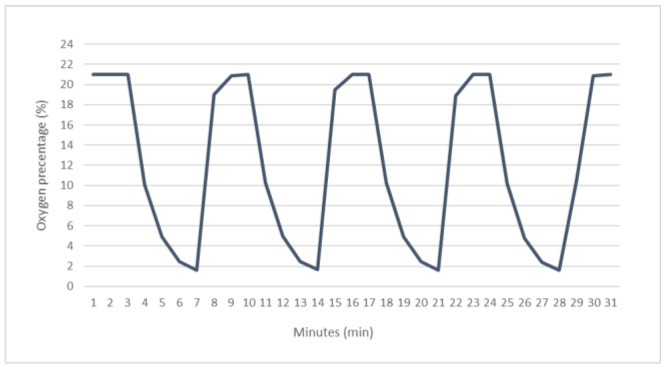
Representative intermittent Hypoxia data measured during a real-time experiment in the OxyCycler incubator chamber. Measurements were collected every 2 s and compressed for representation.

## Data Availability

Data is contained within the article or Appendix A.

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
