# Peer review of "Obstructive Sleep Apnea Syndrome In Vitro Model: Controlled Intermittent Hypoxia Stimulation of Human Stem Cells-Derived Cardiomyocytes"

_ijms, 2022, doi:10.3390/ijms231810272_

Round 1
Reviewer 1 Report
In this manuscript, the authors performed a set of experiments to evaluate several hypoxia-associated parameters in an environment mimicking IH.
With the intent to improve the work done by the authors, I have some suggestions that should be incorporated into the manuscript.
The research question of this experimentation is not clear. What is the scientific gap that this research is aiming to fill?
The link between IH and OSA needs to be better elaborated, and some sort of control/validation for the new proposed equipment used to induce IH should be added.
Immunofluorescence pics are of poor quality/resolution and miss a cardiac myocyte marker. How can readers be sure these cells are cardiac cells?
Figure 1. This review believes that the T-test student is not the appropriate statistical test to be performed when different groups of data are presented in the same graph for comparison
Figure 2. The graph appears to compare different groups and treatments. This review believes that the T-test student is not the appropriate statistical test to be performed for this kind of data comparison.
Figure 3. The graph appears to compare different groups and treatments. This review believes that the T-test student is not the appropriate statistical test to be performed for this kind of data comparison.
Figure 4 misses the statistical analysis.
Author Response
- The research question of this experimentation is not clear. What is the scientific gap that this research is aiming to fill?
Currently there is no good solution to the cardiovascular sequel that occurs in OSA treated by CPAP, this study looks on human cardiomyocytes under hypoxic conditions and aims to identify mechanisms that may become a target for intervention. We search for those mechanisms at the molecular, cellular and physiological levels.
This paragraph was added to the Introduction.
All references mentioned here are also present in the manuscript.
- The link between IH and OSA needs to be better elaborated, and some sort of control/validation for the new proposed equipment used to induce IH should be added.
IH is a key feature of OSA and is used by several groups to study it (Xu et al., 2015). Our system is unique due to the fact we use IH conditions with human CMs, for the first time. In order to validate the effect of IH on these cells, we chose to use Hif-1α as the most familiar marker known in the literature for hypoxia. In figure 2 we show the activation of Hif-1α upon IH on the CMs and thus it validates our system.
- Immunofluorescence pics are of poor quality/resolution and miss a cardiac myocyte marker. How can readers be sure these cells are cardiac cells?
As a known cardiomyocytes marker, we stained with an antibody to cardiac Troponin T (cTnT), (Choi et al., 2021) and were detected by a secondary antibody coupled with a green fluorescence dye. The presence of this green marker represents the cardiomyocytes in figures 1 and 3. The green color is present only in these cells. Cells that are not stained green are not cardiomyocytes and are not scored in our analysis. The pictures are only partially in focus since these are “colonies” of cells which are three dimensional made of several cell layers.
- Figure 1. This review believes that the T-test student is not the appropriate statistical test to be performed when different groups of data are presented in the same graph for comparison.
Following the reviewer’s suggestion, we consulted our statistician and have changed to ANOVA analysis. The data, figures and legends have been updated. Our conclusions after the analysis remained the same.
- Figure 2. The graph appears to compare different groups and treatments. This review believes that the T-test student is not the appropriate statistical test to be performed for this kind of data comparison.
As mentioned in the previous item we have changed to ANOVA analysis. The data, figures and legends have been updated. Our conclusions after the analysis remained the same.
Reviewer 2 Report
This study evaluated the effect of intermittent hypoxia on human stem cell-derived cardiomyocytes, and found up-regulation of Hif-1a and NF-Kb signaling pathways. Although cardiovascular morbidity in OSA patients has been well known, the exact mechanisms leading to these pathological conditions at the cellular and molecular level are unknown. Therefore, although this study is important, it also has several limitations.
Comments
1. In intermittent hypoxia model, alternating condition between normoxia and hypoxia is very important factor. Please explain why 1% for 8 min and 21% for 4 min condition was used for IH.
2. Time or cycle frequency-dependent HIF-1a and NF-kB activation experiments should be performed.
3. Are there data about real-time O2 level measured in cell culture media according to IH condition?
4. HIF-1a and NF-kB signaling activation are also shown in continuous hypoxia condition, thus I recommend comparing the result of the IH protocol with that of the continuous hypoxia condition.
5. I recommend showing the gene expression result of HIF-1a downstream genes (e.g., VEGF, PDK-1) because results of inflammation-associated genes have been represented, in addition to NF-kB activation.
6. Generally, the beating rate in OSA patients varies according to apneic status or normal breathing status. Moreover, several studies consistently reported the sympathetic activation in OSA patients, implying fast heart beating and high blood pressure in the patients. Thus, in my opinion, the decrease in beating rate in cardiomyocytes following IH seems not an important feature.
7. Line 66~67 in the Introduction: “Although CPAP restores respiration and sleep architecture, it does not reduce cardiovascular morbidity”.
Accumulated evidences show the positive effect of CPAP on several cardiovascular morbidities, thus this sentence may be misunderstood, so it is better to delete it.
Author Response
- In intermittent hypoxia model, alternating condition between normoxia and hypoxia is very important factor. Please explain why 1% for 8 min and 21% for 4 min condition was used for IH.
Thank you for this relevant question. Indeed, several researchers studied the effect of continuous hypoxia (intermittent or continuous) on CM as well as on other cell types (Chuang et al., 2016; Murphy et al., 2016; Wu et al., 2016; Xu et al., 2015). The range of O2 % used in the in vitro literature is between 0.0%-3%. Following several trials using higher O2 concentrations, we observed that the best response to IH (Hif-1α activation) in our setting was 1% for 8 min and 21% for 4 min. It is important to note that the time elapsing from hypoxia to recovery to normoxia and back to hypoxia requires several minutes to reach the desired O2%. This representative graph was added to the Materials and Methods section (Figure 5).
All references mentioned here are also present in the manuscript.
- Time or cycle frequency-dependent Hif-1α and NF-kB activation experiments should be performed.
Since this work presents a novel protocol to study the effect of IH on human cardiomyocytes, some of the experiments are planned to show proof of concept, demonstrating a series of effects as compared to continuous normoxic conditions. We agree with the reviewer that in future studies, more detailed experiments, kinetics of time and/or cycle frequency will be performed.
- Are there data about real-time O2level measured in cell culture media according to IH condition?
The O2 levels in the media are not measured during the experiment since a suitable probe is not available to us. The values presented refer to the O2% in the air of the (small) chamber. Since the media volume in each well is also small (0.2 ml), we infer that the values in the medium are close to those in the chamber (See figure 5).
- Hif-1α and NF-κB signaling activation are also shown in continuous hypoxia condition, thus I recommend comparing the result of the IH protocol with that of the continuous hypoxia condition.
Since IH is a key feature of OSA rather than continuous hypoxia, we tried here to mimic the O2 fluctuation in OSA to expand the knowledge of cardiovascular morbidity in patients. We are aware from the literature that continuous hypoxia induces a pro-inflammatory reaction. Here we show as a proof of concept and further validity of our system, that IH has also a pro-inflammatory effect. We agree with the reviewer that in the future, more detailed experiments that will compare between both conditions will be performed.
- I recommend showing the gene expression result of Hif-1α downstream genes (e.g., VEGF, PDK-1) because results of inflammation-associated genes have been represented, in addition to NF-κB activation.
Since the main purpose of determining Hif-1α activation was to validate the IH protocol and not as a detailed study of its effect on gene expression, we did not continue this line of research. We believe this work opens a new avenue to study the effect of IH in human cardiomyocytes and many different lines of detailed studies can be pursued, including the detailed dissection of Hif-1α induced gene expression.
- Generally, the beating rate in OSA patients varies according to apneic status or normal breathing status. Moreover, several studies consistently reported the sympathetic activation in OSA patients, implying fast heart beating and high blood pressure in the patients. Thus, in my opinion, the decrease in beating rate in cardiomyocytes following IH seems not an important feature.
Although sleep is normally a time when parasympathetic modulation of the heart predominates and myocardial electrical stability is enhanced, OSA and CSA disturb this quiescence, creating an autonomic profile in which both profound vagal activity, leading to bradyarrhythmias, and sympatho-excitation favouring ventricular ectopy are observed. The resulting tendency toward cardiac arrhythmia may directly contribute to sudden cardiac death and premature mortality in patients with sleep apnea (Leung, 2009; Taniguchi et al., 2021). Thus, since both a decrease or an increase in beating rate is observed in OSA patients, either variation from the norm, including during apneic episodes can be of significance and important in promoting arrythmias. This paragraph was added in the discussion.
- Line 66~67 in the Introduction: “Although CPAP restores respiration and sleep architecture, it does not reduce cardiovascular morbidity”. Accumulated evidences show the positive effect of CPAP on several cardiovascular morbidities, thus this sentence may be misunderstood, so it is better to delete it.
The prevention of adverse CV events in patients with OSA remains uncertain because several randomized controlled trials (RCTs) and meta-analyses have reported no risk reduction in adverse CV events from the use of CPAP therapy in OSA (Reynor et al., 2022). We based our comment mainly on the work of McEvoy et al published in the New England Journal of Medicine six years ago. After studying more than 2700 adult OSA patients, they found that therapy with CPAP plus usual care, as compared with usual care alone, did not prevent cardiovascular events in patients with moderate-to-severe obstructive sleep apnea and established cardiovascular disease, (McEvoy et al., 2016). Thus, there is now more awareness that CPAP alone cannot prevent OSA cardiovascular complications.
Round 2
Reviewer 1 Report
Thank you for your response and clarification!!
However, Figure 4E still misses the statistical analysis. How can the authors make the comparison between different data without statistics?
Author Response
However, Figure 4E still misses the statistical analysis. How can the authors make the comparison between different data without statistics?
Thank you for the remark, we have clarified this point in the Materials and Methods section:
"In the membranes, each protein is doted in duplicate. The duplicate densitometry values were averaged and compared to the average values obtained in the normoxia membrane, immediately after IH or compared to normoxia after 24 recovery. The results were expressed as percentage of each protein as compared to normoxia. "
Reviewer 2 Report
The manuscript has been well modified according to reviewer's suggestions. I have no other comments.
Author Response
Thanks! we appreciated your comments